# Urban Green Space Composition and Configuration in Functional Land Use Areas in Addis Ababa, Ethiopia, and Their Relationship with Urban Form

**Eyasu Markos Woldesemayat [1,*] and Paolo Vincenzo Genovese [2]**

1    School of Architecture, Tianjin University, Tianjin 300072, China
2    Bionic Architecture and Planning Research Centre, School of Architecture, Tianjin University, Tianjin 300072, China; pavic_genovese@tju.edu.cn
*    Correspondence: eyasumark@gmail.com; Tel.: +86-166-2210-1443 or +251-912630022

**Abstract:** This study aimed to assess the compositions and configurations of the urban green spaces (UGS) in urban functional land use areas in Addis Ababa, Ethiopia. The UGS data were extracted from Landsat 8 (OLI/TIRS) imagery and examined along with ancillary data. The results showed that the high-density mixed residence, medium-density mixed residence, and low-density mixed residence areas contained 16.7%, 8.7%, and 42.6% of the UGS, respectively, and together occupied 67.5% of the total UGS in the study area. Manufacturing and storage, social services, transport, administration, municipal function, and commercial areas contained 11.6%, 8.2%, 6.6%, 3.3%, 1.3%, and 1% of the UGS, respectively, together account for only 32% of the total UGS, indicating that two-third of the UGS were found in residential areas. Further, the results showed that 86.2% of individual UGS measured less than 3000 m$^2$, while 13.8% were greater than 3000 m$^2$, demonstrating a high level of fragmentation. The results also showed that there were strong correlations among landscape metrics, while the relationship between urban form and landscape metrics was moderate. Finally, more studies need to be conducted on the spatial pattern characteristics of UGS using very high-resolution (VHR) images. Additionally, future urban planning, design, and management need to be guided by an understanding of the composition and configuration of the UGS.

**Keywords:** urban green spaces; patches; fragmentation; functional land use; composition and configuration





## 1. Introduction

While the world is facing rapid urbanization and climate change extremes, access to nature is also decreasing, which has been resulting in an increase in exposure to environmental hazards, such as increasing temperatures, urban heat island (UHI) effects, heat waves, and pluvial flooding. Governments around the world have been searching for viable strategies to ameliorate these environmental challenges. One of the important measures to mitigate such challenges is the adoption of urban green spaces (UGS). UGS, which comprise forests, parks, vegetation, grass, and community gardens, are recognized as being vital for a healthy and sustainable living environment because they provide multiple social, economic, and environmental benefits. The benefits, among others, include enriching habitats and biodiversity [1], mitigating UHI effects via shading and evaporation [2], reducing high temperatures and heat waves [3], decreasing the impacts of noise [4], providing carbon storage [5], regulating or attenuating flooding [6], enhancing local resilience and promoting sustainable lifestyles [7], improving mental and physical health [8], fostering social interactions and integration by offering meeting places for residents [9], reducing public infrastructure expenditure, and creating improved opportunities for economic regeneration and significant savings related to healthcare expenditure [1,10]. In line with this, studies have also shown that the ability of UGSs to provide their expected

social, environmental, and economic benefits largely depends on their spatial locations, compositions and configurations [11–34]. Composition refers to the variety and relative abundance of UGS patch types within the landscape, which is typically quantified using the proportions of different land cover types, while the configuration is a measure of the spatial characteristics, such as the arrangement, position, or geometric complexity or layout of the patches [12,13,17]. On the other hand, location denotes spatial accessibility of UGS i.e., spatial locations influences multiple urban ecosystem services provided by UGS [34].

The environmental, economic, and social benefits derived from increasing the composition of UGS in urban areas are well documented in the recent literature, however the ways in which the configurational characteristics of such landscape features influence the benefits are evolving. For instance, Berg [18] identified that there is a strong positive association between increasing the quantity of UGS and perceived mental health and all-cause mortality. Similarly, Jaganmohan and Knapp [13] argued that increasing the proportion of UGS considerably contributes to the mitigation of urban heat islands (UHI) effect. In other words, a high percentage of UGS can effectively consume energy through photosynthesis and transpiration, leading to a lower land surface temperature (LST) [12]. Most importantly, Amani-Beni and Zhang [19] argued that for every 10% increase in the UGS, the LST drops by 0.4 °C, indicating the significance of the UGS composition in providing environmental benefits. In terms of economic rewards, Haaland and van Den Bosch [20] contended that there was a positive correlation between an increase in the UGS quantity and gross domestic product. In addition, Panduro and Veie [21] reported that there is a strong relationship between the size and quantity of UGS and the choice of a place to live, as well as between the UGS and the cost of renting or buying a house.

On the other hand, studies related to UGS configuration have shown that patch density (PD) positively correlates with LST, suggesting that an increase in UGS fragments will result in an increase in LST [14]. Grafius and Corstanje [22] showed that larger UGS facilitate a greater degree of provisioning per area of carbon storage and pollinator abundance than smaller ones. A similar study showed that an increase in UGS area increases the cooling effects of a park, revealing that several small UGS distributed throughout a city do not individually have a great cooling effect on the surroundings [13]. Correspondingly, Xiao and Dong [23] identified that the cooling and humidifying effects of large UGS are more obvious and stable, while the cooling effects of small UGS are more variable. Concurrently, studies have shown that there is a relationship between urban form and UGS spatial pattern characteristics. In cities where buildings are small and arrange in a scattered manner (where the UGS size is relatively higher), the UGS in these surroundings significantly reduce the effects of the LST [24]. In other words, less dense areas do have clear advantages over dense ones in terms of the efficiency of cooling the surrounding environment [25]. In contrast, city centers and industrial areas where the share of UGS is small and the patches are isolated appear to be hostile areas for breeding birds [26] and seem to limit the environmental benefits of UGS.

While there is a growing body of literature on the composition of UGS [14–16,26,27], studies on the spatial pattern configurations of UGS in urban functional land use areas, such as residential, commerce and business, administration, manufacturing and storage, social services, municipal services, and transport infrastructure areas, in the African context are lacking. Earlier studies on UGS in African countries were focused on nature and the challenges of UGS [28,29], depletion or shrinkage of environmental resources [28,30,31], degradation of UGS and the decline of ecosystem services [31–33], and the consequences of the loss of UGS in African metropolitan cities [28], highlighting the fact that studies on the spatial patterns of UGS in urban functional land use areas are non-existent [35]. Even in the global arena, studies on the spatial configuration of UGS and the relationships with social, economic, and environmental benefits are not murky. For instance, the effects of size, shape, and type of UGS on environmental benefits such as cooling remain uncertain [13].

This study aimed to analyze the composition and spatial configuration of UGS in urban functional land use areas in Addis Ababa, Ethiopia. The functional urban land use classes

included in the study were residential (further classified as high-density mixed residence, medium-density mixed residence, and low-density mixed residence areas), manufacturing and storage, social services, transport, administration, commercial, and municipal services. Remote sensing and GIS technology integrated with landscape metrics were utilized to analyze the composition and configuration of UGS in different functional land use areas. Similarly, correlation analysis was utilized to measure the degree of association between the UGS landscape metrics and urban form. In this study, the UGS refers to vegetation, grass, greenways, parks, urban forests, agricultural or crop land, and enclosed garden spaces that are found in built-up and non-built-up functional areas, as stated in the Ethiopian National Urban Green Infrastructure Standard [36].

## 2. Method and Research Materials

### 2.1. Study Area

The study area is Addis Ababa city, which is the capital of Ethiopia. The city is located between 9°0′19.4436″ N and 38°45′48.9996″ E at elevations ranging from 2015 to 3152 m above mean sea level. Its total population as of August 2019 was estimated to be 4,592,000, growing at 3.8% annually [37]. The city is home to an estimated 3.238 million inhabitants and constitutes approximately 20 percent of Ethiopia's urban population [38]. It was established as a nation seat in 1886. It is the major political and economic center of the country, as well as the seat of the African Union (AU). It is divided into ten sub-cities and 116 districts, called woredas.

Although the city was established as the capital of the nation a century ago, it did not exhibit a significant change until the beginning of the 1990s. In 1991, with the change of government, the introduction of a free market economy, and subsequently followed by the implementation of an urban development policy (UDP) in 2005, the city experienced rapid urban growth, meaning the urban landscape has been dramatically transformed (Figure 1). During this period, the city's urban area doubled in size, however accompanied by widespread destruction of UGS [39]. In particular, the implementation of the Integrated Housing Development Program (IHDP) has significantly changed the landscape of the city [40]. Along with the expansion of commercial buildings, industry, micro- and small-scale enterprises, and road infrastructure and social services, this has significantly contributed to the decline of UGS. In effect, per capita UGS areas such as parks have been reduced to below 0.37 m$^2$ [38], which is one of the lowest rates in Africa.

From Figure 1, the total area of Addis Ababa city is 520 km$^2$. This area is classified as built-up and non-built-up. The non-built-up areas generally include areas reserved for environmental functions, while built-up comprises functional urban land use areas. The functional urban land uses included in the study were residential areas (further classified as high-density mixed residence, medium-density mixed residence, and low-density mixed residence), manufacturing and storage, commercial, municipal services, social services, transport, and administration areas, as indicated above. It covers an area of 29,952 ha or 55.7% (more than half of the city urban land) of the entire cityscape (Table 1).

### 2.2. Data and Methods

In this study, we used Landsat-8 Operational Land Imager and Thermal Infrared Sensor (Landsat-8 OLI/TIRS) imagery. A 30 m resolution satellite image covering the entire city area was obtained from the Global Land Cover Facility (GLCF) (Table 2). The image was collected during the dry season to minimize the availability of haze and to obtain cloud-free satellite images of the study area. Sub-setting of the acquired satellite images was carried out to extract the study area from the images by geo-referencing the boundaries of Addis Ababa. It was then projected onto a common coordinate system, the Universal Transverse Mercator (UTM) of WGS84 and Datum Zone 37. Atmospheric and radiometric correction was carried out to remove the influence and eliminate the effects of atmospheric scattering.

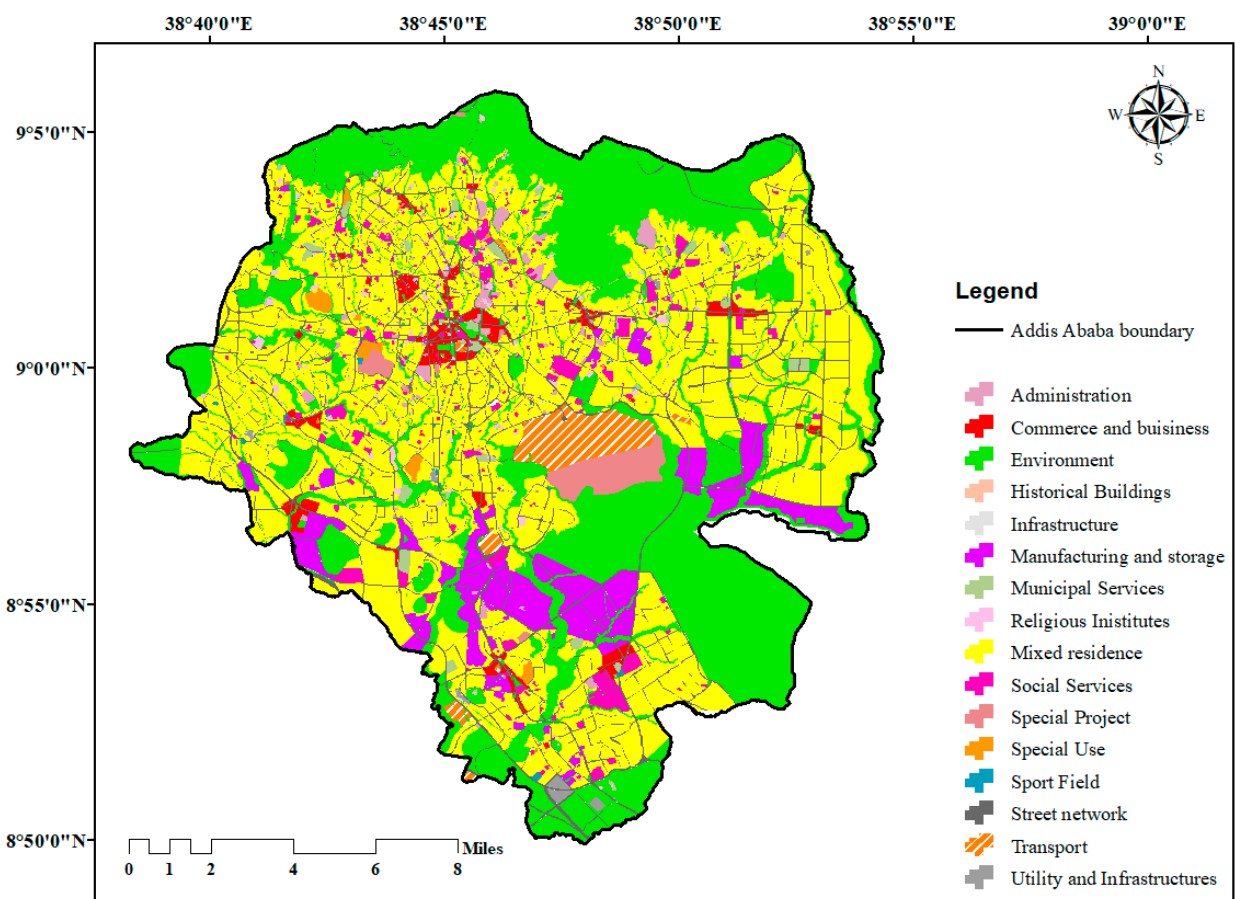

**Figure 1.** Major urban functional land uses in Addis Ababa. Source: Addis Ababa City Planning Commission (AACPC).

**Table 1.** Description of functional land uses included in the study.

| No | Functional Land Use Type | Description |
|---|---|---|
| 1 | Mixed residence<br>1.1 High-density mixed residence.<br>1.2 Medium-density mixed residence.<br>1.3 Low-density mixed residence. | This is the dominant landscape structure in the study area and is further divided into three categories: high-density mixed residence, medium-density mixed residence, and low-density mixed residence, according to the structure plan. |
| 2 | Commercial | Includes landscape that is used for business activities and is heavily paved and dominated by high rise buildings. It covers the inner city core area and along major highways. |
| 3 | Manufacturing and storage | Includes industrial areas, which are largely located in the southern part of the city and used for production and storage. |
| 4 | Municipal services | Municipal service areas comprise areas used for municipal services such as abattoirs, fire and emergency services, green cemeteries, cultural and civic centers, cemeteries, and festival sites and plaza functions. |
| 5 | Social services | Social service areas are built-up areas commonly used for healthcare, stadiums, social care centers, district sports fields, research centers, education, and civic services. |
| 6 | Transport | Transport areas comprise functional spaces designated for bus freight terminals, bus depots, surface parking, parking buildings, intercity terminals, and airports, but excluding linear features such as roads. |
| 7 | Administration | Comprises land use functions mainly used for public services, namely federal institutions, city institutions, sub-city and district administration, as well as inter-governmental organizations. |

**Table 2.** Satellite image source, spatial resolution, and other spatial data sets.

| No | Satellite | Sensor ID | Resolution | Date Acquired | Path and Raw | Source |
|---|---|---|---|---|---|---|
| 1 | Landsat 8 | OLI_TIRS | 30 m * 30 m | 20 December 2016 | 169/055 | Global Land Cover Facility (GLCF) |
| 2 | | | Addis Ababa boundary (shape file format) | | | City Planning Commission |
| 3 | | | Structure plan of Addis Ababa (shape file format) | | | City Planning Commission |
| 4 | | | Building foot print of Addis Ababa city (DWG file) | | | Integrated Land Information Center |

In the present study, to classify the satellite image, a stratified random distribution method was used to generate a total of 300 points in the data set. The validity at each point was cross-checked using Google Earth™, the latest structure plan of the city, a cadastral map of the city, and knowledge of the study area. A visual interpretation approach was also applied to distinguish the image characteristics based on the differences in color, tone, texture, and shape, subsequently followed by maximum likelihood classification (MLC), which is a supervised classification method to obtain six land use and land cover (LULC) classes. The identified LULC classes were agriculture, bare soil, built-up, urban forest, vegetation/grass, and water body. They are generally categorized as built-up or non-built-up areas. The non-built-up areas included agricultural land, vegetation/grass, and urban forest classes, which were reclassified as UGS of the study area. The no-built up areas were excluded from the study because the aim was understanding the composition and configuration of UGS in functional urban land uses areas. Similarly, error matrix, Kappa statistics, and overall accuracy reports were generated from ERDAS IMAGINE® 2014.

To obtain UGS fragments of the functional land uses, the UGS vector data extracted from satellite imagery were over laid with land use spatial data obtained from the AACPC in the ArcGIS 10.4 environment. The result were summarized according to the functional land use areas. Further, to maintain consistency in the analysis of the spatial character tics of the UGS fragments, they were clustered into areas <3000 m², 3000–10,000 m², and >10,000 m² based on the hierarchy of UGS typologies (such as parks), as recommended by the structure plan for Addis Ababa city [41]. The structure plan of Addis Ababa stipulates each neighborhood to have 3000 m² park with in 300 m.

### 2.2.1. Normalized Difference Vegetation Index (NDVI) Calculation

The normalized difference vegetation index (NDVI), which is an index of plant "greenness" or photosynthetic activity, is one of the most commonly used vegetation indices. Vegetation indices are based on the observation of different surfaces that reflect different types of light differently. Photosynthetically active vegetation, in particular, absorbs most of the red light that hits it while reflecting much of the near-infrared light. Clearly, vegetation that is dead or stressed reflects more red light and less near-infrared light. Likewise, non-vegetated surfaces show much more even reflectance across the light spectrum. NDVI is commonly calculated on a per pixel basis as the normalized difference between the red and near-infrared bands from an image and is computed as follows:

$$NDVI = NIR - RED/NIR + RED \tag{1}$$

where NIR is the near-infrared band value for a cell and RED is the red band value for a cell. Calculations of NDVI for a given pixel always result in a number that ranges from minus one ($-1$) to plus one ($+1$); however, an absence of green leaves will give a value close to zero. A zero means no vegetation, while close to $+1$ (0.8–0.9) indicates the highest possible density of green leaves.

### 2.2.2. Landscape Pattern Analysis

Landscape pattern metrics are widely used to measure the spatial composition and configuration of UGS, because of their ability to describe complex characteristics of patches, such as the patch shape, size, quantity, and spatial combination. However, the selection of

landscape pattern metrics for a study depends on the application, as some of the indices exhibit similarities and overlap extensively [42]. Hence, the suitability of landscape pattern metrics needs to be evaluated, as suggested by [43–45]. In this study, we used the most commonly applied compositional and configurational metrics in recent research studies, and the metrics were generally classified as compositional or configurational (Table 3). Compositional metrics provide evidence of the quantity of the UGS, whereas configurational metrics offer information on the geometrical characteristics of the UGS patches.

**Table 3.** Description of the spatial metrics used for the study (McGarigal et al., 2002).

| Metric Type | Metric | Abbreviation | Description | Range | Unit |
|---|---|---|---|---|---|
| Compositional | Class area | CA | The sum of all patches for the corresponding patch type. | $CA \geq 0$ | Hectares |
| | Number of patches | NP | NP equals the number of patches for the corresponding patch type (class). | $NP \geq 1$ | None |
| | Largest patch index | LPI | Equals the area ($m^2$) of the largest patch in the landscape divided by the total landscape area ($m^2$). | $0 < LPI \leq 100$ | Percent |
| Configurational | Mean shape index | SHAPE_MN | Average shape index for the corresponding patches within an analysis unit. | $AREA\_MN \geq 0$ | None |
| | Mean Euclidean nearest neighbor distance | ENN_MN | Refers to mean distance to the nearest neighborhood patch of urban UGS based on the edge-to-edge distance | $ENN\_MN > 0$ | m |
| | Patch density | PD | Measures the density of patches for each class in the entire landscape. | Number of patches per 100 ha | $PD > 0$ |

The landscape metrics used in the present study were the total area (CA), number of patches (NP), patch density (PD), largest patch index (LPI), mean patch shape index (SHAPE_MN), and Euclidean nearest neighbor distance (ENN_MN). The metrics were computed at class the level in FRAGSTATS 4.1.1 software, as indicated above.

### 2.2.3. Urban Form Metrics

A growing number of studies have shown that urban form influences the spatial pattern characteristics of the UGS in urban areas [42,46–48]. For instance, Huang and Yang [46] utilized topography, Perimeter Area Ratio (PARA), and road network density (RD) to analyze the correlation between UGS and urban form. Simwanda and Ranagalage [47] attempted to examine the relationships between the LST, composition, and configuration of impervious surfaces in four African cities, namely Addis Ababa, Nairobi, Lusaka, and Lagos. Davies and Barbosa [49] used the building coverage, length of a road, housing density, elevation, and slope to understand how urban form influences the availability of the UGS. Tratalos and Fuller [50] studied the relationship between urban form and the ecosystem performance of UGS in five UK cities. In this study, we used three urban form indicators, namely the building coverage (BC), perimeter–area ratio (PARA), and building density (BD) to comprehend the relationships between urban form and the UGS composition and configuration.

Building coverage (BC) refers to an area occupied by a building in an urban area and is one factor that influences the composition and configuration of UGS. It is calculated as the total building area (BA) divided by the total area (TA), as suggested by [51,52] and computed as follows:

$$\text{Building coverage (BC)} = \text{BA/TA} \tag{2}$$

where BA is the building footprint for functional land uses, TA is the total area of land use types, and BC is the building coverage. The BC is represented as a ratio.

Similarly, perimeter–area (PARA) ratio is also an important indicator of the urban form. The PARA ratio shows the shape complexity of each urban area. This was calculated by using the perimeter (P) of a building's footprint over the area (A) of the same building.

$$PARA = Perimeter\ (P)/Area\ (A) \tag{3}$$

In addition, population density, which is the number of people per unit area, was used to measure the relationships between population distribution, UGS composition and configuration. To quantify the population density, the latest projection for the year 2017 was collected from the central statistical authority (CSA). However, this projection was only for the city and sub-cities, and thus it was assumed that the population growth trend followed the same magnitude in all districts.

$$PD = Population\ (P)/Area\ (A) \tag{4}$$

where PD is the population density, P is the total population of the area, and A is the total area, which can be expressed either as ha or m$^2$.

Similarly, building density (BD) is also an important measure of urban form, as high building densities reduce the adverse environmental, social, and economic costs of urbanization [51]. This is computed as the number of building units per unit area (e.g., buildings per hectare) [53]:

$$BD = Number\ of\ buildings\ (NB)/Area\ (A) \tag{5}$$

where BD is the building density, NB refers to the total number of buildings in a given area, while A represents the total area. Finally, computation of urban form indices such as PARA, BC, and BD was carried out in the ArcGIS software 10.4 version platform by using building footprint spatial data obtained from Addis Ababa City Administration Integrated Land Information Center (AAILIC).

### 2.2.4. Statistical Analysis

Statistical analysis was performed to evaluate how landscape pattern metrics and urban form related to each other for the different functional land use types. A Pearson correlation analysis was used to identify moderate, weak, strong, and no correlation among UGS indices and urban forms in each functional land use class. The relationship was assessed using bivariate analysis in SPSS 2023 statistical software.

## 3. Results

### 3.1. Statistical Analysis Result

The correlation analysis result showed that landscape metrics were strongly correlated with each other. For instance, CA exhibited a strong positive correlation with NP (correlation = 0.99 **, $p < 0.001$), while PD was strongly negatively correlated with SHAPE_MN (correlation = −0.93 **, $p > 0.001$) (Table 4). Similarly, the nearest neighborhood distance (ENN_MN) negatively correlated with the CA (correlation = −0.61), while it was positively related with LPI. The results also suggest that an increase in PD results in an increase in shape complexities and reduces the connection of the UGS patches. In contrast, a decrease in PD results in decrease in shape complexities and increases in connection of patches.

Similarly, the degree of association among urban form indices and landscape metrics showed moderate correlations. As can be seen from Table 4, BC was moderately negatively correlated with LPI (correlation = −0.57), while it was fairly positively correlated with PD (correlation = 0.57), implying that an increase of BD increases the complexity of UGS patches. Conversely, BD moderately negatively correlated with LPI (correlation = −0.54), indicating that an increase in BD reduces UGS patch size. Similarly, SHAPE_MN was negatively correlated with BD (correlation = −536), while PD was positively associated with BC (correlation = 570). The relationship between PARA and landscape pattern metrics

was also found to be very weak (Table 4), presenting that changes in the perimeter–area ratio may not significantly affect the composition of the UGS. Similarly, BC is negatively correlated with LPI (correlation = −0.560). The correlation between BD and PD was also identified as milder (correlation = 458) and between BD and NP was fair (correlation = 475), indicating that the effects of urban form on the configuration and composition of the UGS were not strong.

**Table 4.** Correlations analysis results for landscape metrics and urban form indicators.

|   |          | 1       | 2     | 3       | 4     | 5     | 6     | 7     | 8    | 9    |
|---|----------|---------|-------|---------|-------|-------|-------|-------|------|------|
| 1 | CA       | 1.00    |       |         |       |       |       |       |      |      |
| 2 | NP       | 0.99 ** | 1.00  |         |       |       |       |       |      |      |
| 3 | PD       | −0.03   | 0.10  | 1.00    |       |       |       |       |      |      |
| 4 | LPI      | −0.58   | −0.62 | −0.40   | 1.00  |       |       |       |      |      |
| 5 | SHAPE_MN | 0.04    | −0.09 | −0.93 **| 0.26  | 1.00  |       |       |      |      |
| 6 | ENN_MN   | −0.61   | −0.57 | 0.27    | 0.63  | −0.46 | 1.00  |       |      |      |
| 7 | PARA     | −0.07   | −0.02 | 0.27    | 0.36  | −0.37 | 0.33  | 1.00  |      |      |
| 8 | BC       | 0.03    | 0.10  | 0.57    | −0.57 | −0.30 | −0.37 | −0.32 | 1.00 |      |
| 9 | BD       | 0.42    | 0.47  | 0.46    | −0.41 | −0.54 | −0.07 | −0.18 | 0.45 | 1.00 |

**. Correlation is significant at the 0.01 level (two-tailed).

As population distribution one of important factor that determines the spatial composition and configuration of the UGS, it was computed to measure how it influences the UGS spatial pattern characteristics. The result also showed that the influence of population density on the spatial composition and configuration of the UGS was discernable. Specifically, it was identified that the population density and number of fragments were moderately negatively correlated (correlation = −0.581), while the population density and area of the UGS were also fairly correlated (correlation = −0.530), showing that population density was one of the factors that influenced the availability of the UGS.

*3.2. Urban Green Space Distribution in Addis Ababa*

The results of this study showed that the UGS in Addis Ababa is characterized by wide spatial coverage and consists of small isolated and clustered patches. As can be seen in Figure 2, the UGS were dispersed and covered a larger area, revealing a high level of fragmentation. In terms of abundance, visual observation of Figure 2 shows that inner city areas are undersupplied with UGS, while outer areas were better supplied with the UGS. Directionally, the abundance of UGS is relatively better in the northern part of the city, where the area is reserved for multifunctional forests (environmental functions).

The relatively high UGS composition in the northern area was attributed to the presence of a hundred-year-old Eucalyptus tree forest and the terrain characteristics of the area. Unlike other areas, the UGS in these areas displayed relatively a high degree of contagions, with proximity to the neighborhood patches being minimal. On the other hand, compared with the UGS patches in the northern part of the city, the UGS patches in the southern and eastern parts of the city were very small in size and the distances among these patches were larger.

*3.3. Urban Green Space Distribution and Population Density*

Numerous earlier studies have revealed that the distribution of UGS correlates with the spatial distribution of a population, i.e., where there are agglomerations of people and economies, the composition of the UGS is low. In line with this, the study showed that the composition of UGS was influenced by the population distribution of the study area. As can be seen in Figure 3, in areas where there is high population density, there is a low proportion

of UGS. Importantly, the analysis revealed that inner sub-cities had per capita UGS levels below 1 m$^2$, showing that population density influenced the composition of the UGS.

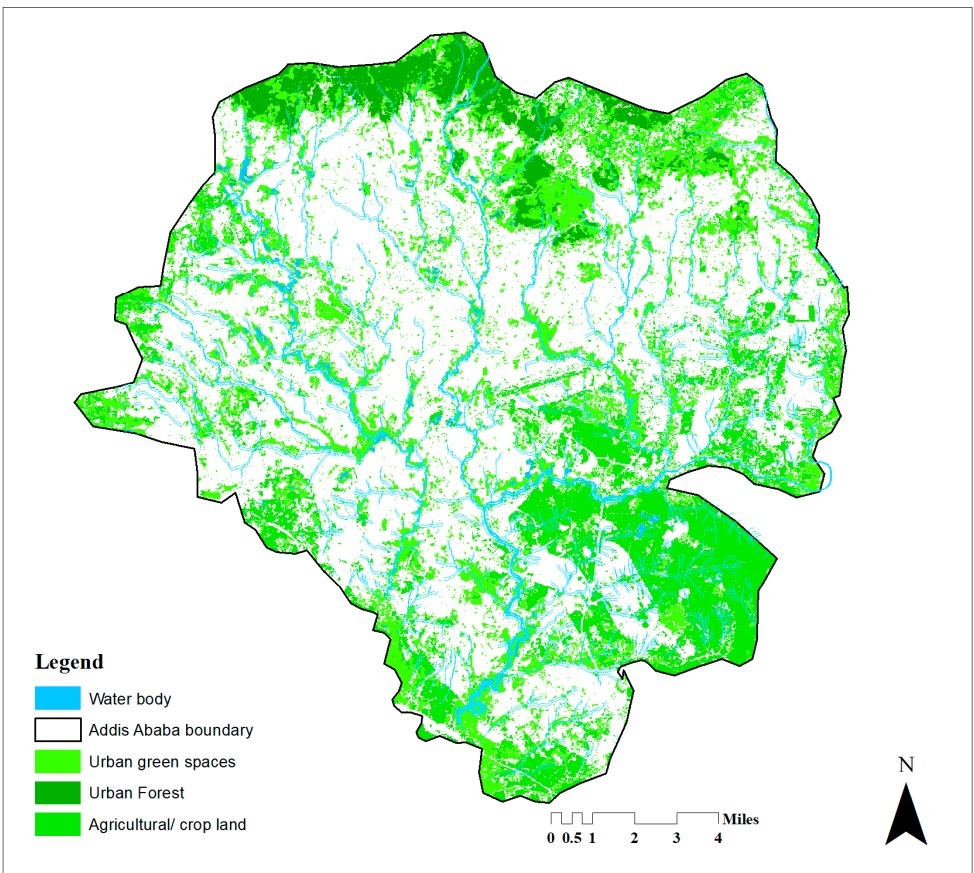

**Figure 2.** Urban green space (UGS) distribution in Addis Ababa, 2020.

Similarly, as can be seen in Figure 3, the per capita level ranges from less than 1 m$^2$/inhabitant (the lowest) in Addis Ketema (one of the inner sub-cities of Addis Ababa) to 102.6 m$^2$/inhabitant in Akaki Kaliti (the highest) (one of the outer sub-cities of Addis Ababa), averaging 28.6 m$^2$/inhabitants. Clearly, the lowest UGS per capita was observed in the oldest inner city area, where the density was the highest (44,815 inhabitants/km$^2$). According to the latest structure plan and master plan for the city, these areas are used for commerce and business activities and were significantly lacking of UGS.

Correspondingly, the per capita distribution among the districts in the study area also provided supporting evidence of the inequalities of the UGS supply. As can be seen in Figure 3, of the total number of 116 districts in the city, 61 districts reported per capita UGS of less than 15 m$^2$/inhabitant (below the Ethiopian Urban Green Infrastructure standard), demonstrating that more than 50% of the districts were poor in terms of the availability of UGS, and thus require urgent urban planning intervention to increase the quantity of UGS (Table 5). The high proportion of per capita UGS in the peripheries is related to the presence of agricultural or crop land, expansion areas with UGS land covers and multifunctional forests, which are classified as environmental function in the latest structure plan of the city.

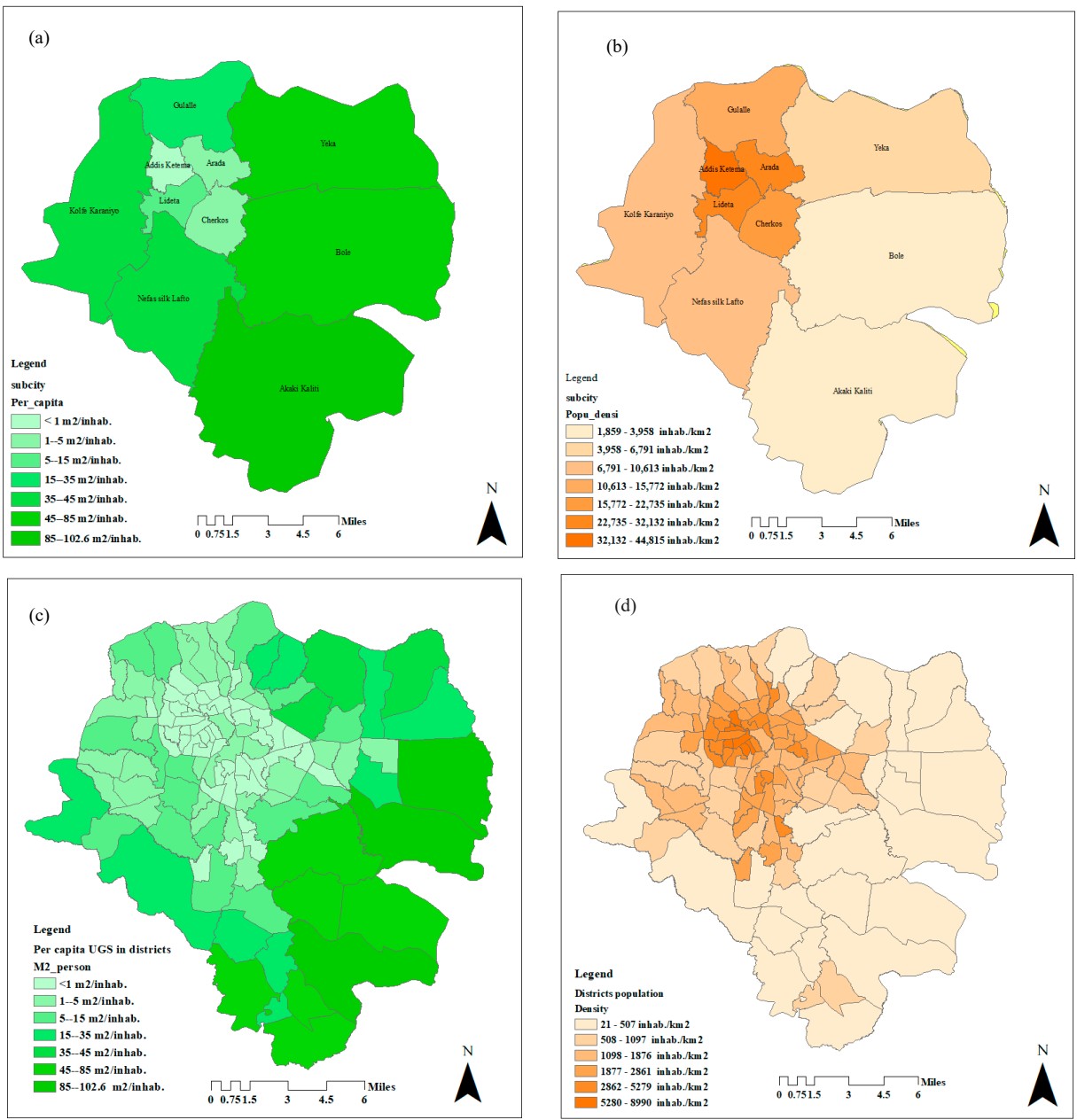

**Figure 3.** Population density and per capita UGS in Addis Ababa: (**a**) urban green space per capita in the sub-cities; (**b**) population density in the sub-cities; (**c**) urban green space per capita in the districts; (**d**) population density in the districts. Note: inhab. means inhabitants.

**Table 5.** Urban green space per capita distribution in the districts in Addis Ababa.

| Per Capita UGS | No of Districts | % of Districts | Remarks |
|---|---|---|---|
| <1 m$^2$/inhabitant | 10 | 9 | |
| 1–5 m$^2$/inhabitant | 23 | 20 | UGS 15 m$^2$/inhabitant is the |
| 5–10 m$^2$/inhabitant | 14 | 12 | local standard and 9 |
| 10–15 m$^2$/inhabitant | 14 | 12 | m$^2$/inhabitant is the WHO |
| 15–30 m$^2$/inhabitant | 16 | 14 | standard |
| 30–705 m$^2$/inhabitant | 39 | 34 | |
| Total | 116 | 100 | |

### 3.4. Spatial Distribution of Urban Green Spaces in Urban Functional Land Use Areas

Spatial analysis showed that Addis Ababa city had a total coverage area of 9700 ha UGS. Of this, the functional land use areas included in this study covered only 919 ha (9.4%), implying that a larger proportion of the UGS was found outside the main functional land use areas. Meanwhile, in terms of the proportion of the UGS in functional land use areas, the results showed that the residential areas had higher UGS coverage compared to other functional use areas (Table 6). In particular, the residential areas classified as a high-density mixed residence, medium-density mixed residence, and low-density mixed residence contained 16.7%, 8.7%, and 42.6% of the UGS, respectively. Overall, these areas contained 68% of UGS in the study area, while manufacturing and storage shared the second largest proportion of the UGS (11.5%) in the study area. The remaining functional uses such as transport, social services, municipal services, and commercial areas and business accounted for 8.2%, 6.6%, 1.3%, and 1.7%, respectively, overall sharing only 17.8% of the total UGS area in the study area.

**Table 6.** Functional land use and urban green space proportions in Addis Ababa.

| No | Functional Land Use Type | Land Area (ha) | Land (%) | UGS Area (ha) | UGS (%) |
|---|---|---|---|---|---|
| | Mixed residence | | | | |
| 1 | 1.1. High-density mixed residence | 2628.96 | 8.8 | 153.3 | 16.7 |
| | 1.2. Medium-density mixed residence | 3921.98 | 13.1 | 80 | 8.7 |
| | 1.3. Low-density mixed residence | 15,481.3 | 51.7 | 392 | 42.6 |
| 2 | Commerce and business | 760.1 | 2.52 | 9.4 | 1 |
| 3 | Municipal services | 589.7 | 2 | 12.1 | 1.3 |
| 4 | Social services | 1483.60 | 5 | 61.1 | 6.6 |
| 5 | Transport | 1152.60 | 3.82 | 75.8 | 8.2 |
| 6 | Manufacturing and storage | 3099.50 | 10.3 | 105.6 | 11.6 |
| 7 | Administration | 834.7 | 2.8 | 30.4 | 3.3 |
| | Total | 29,952.44 | 100 | 919.7 | 100 |

The reason for such a high proportion of the UGS being in the low-density mixed residence areas in particular and residential areas in general is attributed to the size of the residential areas. As can be seen in Table 6, for instance, the low-density mixed residence class comprised an area three times larger than the total area of the high-density mixed residence and medium-density mixed residence classes, implying that the proportion was related to the spatial size of the functional land use areas.

The study also showed that there were 3881 UGS fragments in the functional urban land use areas (Table 7). Of these fragments, 3348 (86.2%) fragments had an area of less than 3000 m$^2$, 424 (10.9%) had a size between 3000 and 10,000 m$^2$, and 109 (2.9%) had an area greater than 1 ha, revealing that the UGS in the functional use areas were small in size and highly fragmented. Most importantly, one of the key findings of this study is that in commercial functional land use areas there were no UGS patch which had area greater than 10,000 m$^2$, indicating that the UGS in this functional land use area was very small compared to others. In terms of the average area of the UGS patches, commercial area UGS patches had (1,377.3 m$^2$) and municipal function area UGS patches had (1,891.6 m$^2$) reporting very small sizes, a size less than neighborhood park size recommended by the structure plan of the city, indicating high fragmentation. Overall, the UGS fragmentation was high in Addis Ababa and the spatial pattern was mainly characterized by small, isolated, scattered UGS patches lacking connectivity.

**Table 7.** Number of fragments and their proportions in the functional urban land use areas in the study.

| | Urban Green Spaces Patches | | | | | | | |
|---|---|---|---|---|---|---|---|---|
| | Residential | Commercial | Administration | Manufacturing | Social services | Municipal services | Transport | Total |
| Number | 2812 | 68 | 115 | 375 | 272 | 64 | 175 | 3881 |
| Percentage | 72.4 | 1.75 | 2.96 | 9.66 | 7 | 1.64 | 4.5 | 100 |
| <3000 m$^2$ | 2445 | 61 | 100 | 304 | 234 | 55 | 149 | 3348 |
| (3000–10000 m$^2$) | 311 | 7 | 9 | 46 | 29 | 8 | 14 | 424 |
| >10,000 m$^2$ | 56 | 0 | 6 | 25 | 9 | 1 | 12 | 109 |
| Average area (m$^2$) | 1804.4 | 1377.3 | 2639.9 | 2816.2 | 2246.5 | 1891.6 | 4329.9 | |

*3.5. Landscape Metrics Analysis*

The result of the landscape metrics analysis for the UGS for seven land use classes showed variation in their spatial patterns, compositions, and configurations (Table 8). For instance, the number of patches (NP) was markedly varied in the functional urban land use areas and that patches were located in residential areas. Incongruent with this, landscape metrics analyses revealed that there were 6514 patches in the residential areas, which accounted for 88.5% of the total number of patches in the study area. Specifically, the numbers of patches in high-density residence, medium-density residence, and low-level residence areas were 447, 436, and 1731, respectively, indicating that the largest number of patches was found in low-density mixed residence areas. Similarly, commercial areas and municipal function areas had the lowest numbers of patches (both 58), which may have been due to the fact that municipal functional land use areas such as cemeteries in Addis Ababa lack vegetation [54]. The patch density (PD), which is a measure of the spatial pattern characteristics of the UGS in a given area, also revealed similar results. The PD values for commercial, medium-density, and municipal service areas were 541.6, 460.9, and 438.4, respectively. The high PD values mean that the UGS in these areas is fragmented into smaller patches. Meanwhile, values for the largest patch index (LPI), which quantifies the percentage of total landscape area, varied according to the land use classes. For example, transport and municipal service classes had the highest LPI values (25.1 and 14.3, respectively), while the low-density residential areas reported the lowest LPI value (1.8), indicating that transport infrastructure areas such as Bole International Airport contain large UGS patches, contrary to commercial function areas in Addis Ababa city.

The results showed that the shape complexity of UGS patches marginally varied according to the land use types. The SHAPE_MN values for transport, manufacturing and storage, administration, and high-density residence areas reflected high complexity (1.15, 1.13, 1.13, and 1.12, respectively), while commercial and municipal service areas showed relatively low complexity (1.08 and 1.07, respectively). The ENN_MN results, which is an indicator of the spatial configuration of the UGS, provided insight into the spatial proximity of the UGS fragments.

The ENN_MN values for municipal service and commercial areas were 506 m and 362.8 m, respectively, while the value for the administrative area was 327.9 m, indicating that the UGS in these land use areas were scattered and spatially isolated from each other. On the other hand, the ENN_MN values for residential areas were 149.9 m, 155.5 m, and 142.1 m for high-density mixed residence, medium-density mixed residence, and low-density mixed residence areas, respectively, revealing that UGS patches located in these functional urban land use areas were close to each compared to commercial, municipal, and administration function areas.

**Table 8.** Landscape metrics and urban form analysis results.

| Functional Land Use Types | CA | NP | PD | LPI | SHAPE_MN | ENN_MN | PARA | BC | BD |
|---|---|---|---|---|---|---|---|---|---|
| 1. Mixed residential class | | | | | | | | | |
| 1.1. High-density mixed residence | 142.8 | 447 | 313 | 2.8 | 1.13 | 149.9 | 0.342 | 0.336 | 203 |
| 1.2. Medium-density mixed residence | 94.6 | 436 | 460.9 | 2.6 | 1.09 | 155.5 | 0.86 | 0.36 | 211 |
| 1.3. Low-density mixed residence | 445 | 1731 | 389 | 1.8 | 1.1 | 142.1 | 0.873 | 0.146 | 217 |
| 2. Commercial Class | 10.7 | 58 | 541.6 | 14.3 | 1.08 | 362.8 | 0.975 | 0.309 | 154 |
| 3. Administration Class | 32.8 | 94 | 286.9 | 17 | 1.13 | 327.9 | 0.769 | 0.122 | 86 |
| 4. Manufacturing Class | 111.8 | 310 | 277.3 | 6 | 1.13 | 162.6 | 0.814 | 0.08 | 68 |
| 5. Municipal services | 13.2 | 58 | 438.4 | 14.3 | 1.07 | 506 | 0.832 | 0.039 | 198 |
| 6. Social services | 64.7 | 219 | 338.4 | 11.1 | 1.12 | 256.9 | 0.826 | 0.125 | 110 |
| 7. Transport | 72.7 | 103 | 141.6 | 25.1 | 1.15 | 260.7 | 0.882 | 0.02 | 141 |
| Total | 988.3 | 3456 | | | | | | | |

## 4. Discussion

### 4.1. Urban Green Space Distribution in Addis Ababa

This study has shown that the distribution of UGS in Addis Ababa markedly varied across the landscape. In terms of abundance, the UGS coverage was very low in the southern, eastern, and western parts of the city, while UGS coverage was relatively high in the northern part of the city. The high level of UGS in the northern part of the city is attributed to the terrain characteristics of the area. The terrain is a physical barrier for urban expansion and contributed to a relatively high proportion of the UGS, a finding that agrees with [55]. According to Qian and Wu [55], urban expansion occurs mainly in relatively low-altitude areas, and thus elevations are a constant barrier for expansion. On the other hand, the UGS coverage in the inner city (oldest city neighborhoods) areas was very low compared to peripheral areas. In Addis Ababa, inner city areas are located in the urban core, where low-income residents live in neighborhoods undersupplied with basics amenities such as UGS. These areas had a very low proportion of the UGS patches, which has been investigated in many earlier studies carried out in other countries [20,56]. Haaland and van Den Bosch [20] showed that most central districts are dominated by commercial land and in these areas space is very limited due to urban densification and the UGS generally being small, irregular, scattered, and fragmented. In line with this, Soltanifard and Aliabadi [57] showed that inner urban areas have less open spaces and green vegetation cover compared to others. However, studies conducted in Asian city such as Beijing, China showed that older city areas had high proportions of UGS. For instance, according to Sun and Lin [58], older city areas in Beijing, which were built 106 years ago, had a greater proportion of UGS, containing 73.03% of the UGS in the city, while the areas built 10 years ago had the smallest proportion of UGS (31.6%), highlighting the role of restrictive urban planning policies implemented in recent years. Conversely, although majority of inner city areas built long ago, they are obsolete, congested and lack UGS, because they were incongruently planned and exhibited unplanned spatial growth over years. In the past two decades, while the city has been expanding outward, a compact urban development policy approach has also been implemented in the inner part of the city, so as to make the city competitive for business and introduce new urban planning standards. Even though there are positive outcomes of this effort, allocation of adequate UGS spaces has been largely overlooked and ineffective in terms of meeting minimum standard. This observation is different in Asian cities such as Xiamen, China. In Xiamen, despite the compact urban development, newer built-up spaces occupy a larger proportion of the total area and the new UGS, are diverse, and provide larger per capita compared to other areas [58].

Similarly, this study has shown that the districts in Addis Ababa have an uneven distribution of per capita UGS. Inner districts had low per capita UGS levels, while outer districts had higher per capita UGS levels. Inadequate supply of UGS means that the per capita UGS level is low and this has been reported in many developing countries cities and comparable with earlier findings [33,59]. For example, according to the World Bank [59], the share of all parks, recreation areas, greenways, waterways, and other protected areas accessible to the public is estimated to be below 1 m$^2$ per inhabitant in cities such as Luanda, Cairo, and Alexandria, i.e., far below the 9 m$^2$ and 30 m$^2$ per capita recommended by the World Health Organization and the United Nations, respectively.

*4.2. Urban Green Space Composition in Functional Land Use Areas*

This study identified that there was variation in the composition and spatial configuration of the UGS in the seven functional land use areas. Among others, the residential area in general and the low-density mixed residential areas in particular had high UGS coverage compared to other areas. The variation in the composition of the UGS between land use classes may be associated with the spatial locations of low-density areas, the differences in their levels of anthropogenic activities (people tend to plant trees for various reasons), the patterns of settlement, and the spatial heterogeneity of urban functional land uses, correlating with previous studies [32,60]. For instance, Chimaimba and Kafumbata [60], in their study on urban tree species composition and diversity in Zomba city, Malawi, identified that variation in species composition between land use types was associated with the differences in their levels of human activities, i.e., people in urban areas tend to plant trees. Consistent with this, Sun and Lin [58] identified that UGS in residential areas in Beijing, China, accounted for 44.8% of all urban UGS, showing the influence of urban land use functions on the composition of the UGS. A similar study also showed that tree canopies were abundant in residential areas than other functional land use areas in Boston, USA [61]. A high proportion of UGS in residential areas (especially in low-density mixed residence) may also be associated with the spatial locations of these areas in developing countries. In many African cities, low density settlements are found in the outer zones of the cities and in these areas the UGS is generally high. Arguably, Lindley [32], in their most recent study, showed that low-density areas are located in the outer areas of Addis Ababa and have a high proportion of UGS compared to the inner city areas. Similarly, Andersson and Haase [26] indicated that low-density urban settings have more built UGS compared to high-density areas. On the other hand, medium-density mixed residence, high-density mixed residence, and commercial functional land use areas had lower UGS coverage compared to low-density mixed residence areas, which may be due to the fact that the areas were more covered by built surfaces such as buildings, roads, and other urban infrastructure. This result is comparable with those shown by Xlu and Ignatieva [62], who investigated whether high-density settlements are characterized by more built surfaces and are devoid of UGS. However, studies have shown that the composition of UGS not only varies with land use type, but also varies according to socioeconomic status. People with high socioeconomic status have better access to more UGS or larger areas of green space than people low socioeconomic status [63,64]. McConnachie and Shackleton [63], in their study on the distribution of public UGS in 9 small towns in South Africa, found that there are significant differences between the amounts of UGS in poor and affluent areas; in particular, the black South African areas are poorly endowed with UGS. Importantly, Venter and Shackleton [65] showed that white residents (who have 6-fold higher income than others) had 11.75% greater tree coverage, 8.9% higher vegetation greenness coverage, and live 700 m closer to a public park than areas with predominately black African, Indian, and colored residents live, highlighting inequalities in neighborhood greenness levels related to income and race.

### 4.3. Urban Green Spaces and Population Density

The results also revealed that the per capita UGS strongly correlated with population density. A logarithmic equation fitted to the proportion of UGS, the number of fragments, and the population density demonstrated that there are strong relationships among them. As indicated in Figure 4 below, an increase in the population density leads to a reduction of the UGS area, reporting a positive correlation shown between them. The strength of the correlation between the population density and number of fragments was slightly stronger (R = 0.686) than between the population density and the UGS area (R = 0.517). However, the literature presents diverging views on the relationship between density and the proportion of UGS. Kabisch and Hasse [66], in their study on green spaces in European cities revisited over the period 1990–2006, showed that there was no correlation between per capita UGS and population density, showing that a decrease in the population does not necessarily lead to more UGS. Silva and Viegas [67], in their studies on environmental justice and inaccessibility to green infrastructure in two European cities, identified that the level of UGS in Tartu, Estonia, at 20.8 $m^2$/inhabitant, was higher than in Faro, Portugal, at 6.2 $m^2$/inhabitant. The high proportion of UGS in Tartu was attributed to the low density of the development in the city [67]. Correspondingly, Shekhar and Aryal [68] argued that there is a negative correlation between population density and per capita green space, i.e., high-density wards have less per capita UGS, while outer wards have relatively high per capita UGS, because of the presence of low-density settlement. The results are also in agreement with studies conducted in Hong Kong and Singapore. In Hong Kong, as a result of the presence of high-density settlements and very limited open spaces, there was less green space in some areas [69]. In general, old inner city areas had very low UGS coverage as compared to the other areas, with densely populated urban areas experiencing the greatest loss of greenness [70].

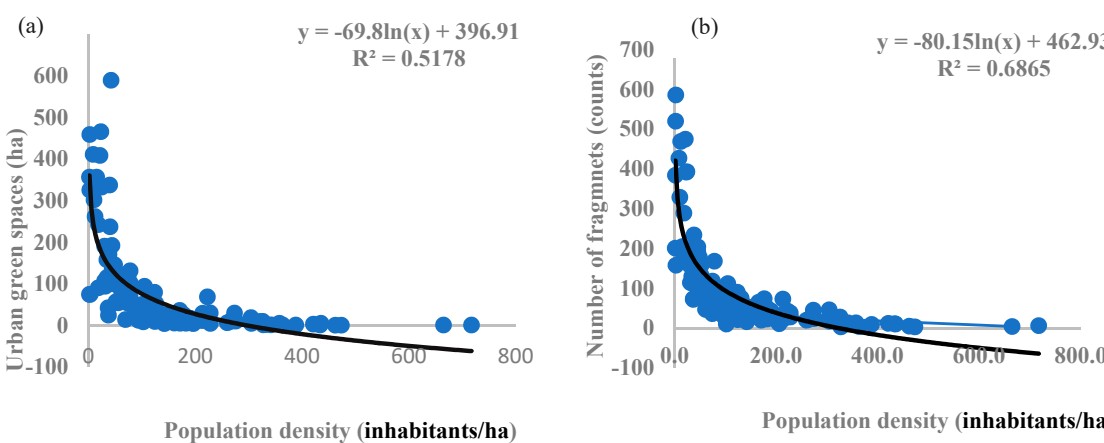

**Figure 4.** Correlations among urban green spaces, urban green space fragments, and population density in Addis Ababa: (**a**) correlation between urban green spaces and population density; (**b**) correlation between number of fragments and population density.

### 4.4. Urban Green Spaces, Urban Form, and Landscape Pattern Indices

The landscape metrics results showed that the UGS patches varied compositionally and configurationally across the seven functional land use classes. Among the seven land use classes, the NP values in residential functional land use areas were the highest, especially in low-density residential areas. The high NP in low-density mixed residential areas relates to the pattern of urban growth in Addis Ababa. Urban growth in Addis Ababa is characterized by large scale informal settlements expansion like many developing country cities in Africa and accounts for more than 60% of the total area [29,31,34]. Mpofu [29], in his study on the environmental challenges of urbanization in Addis Ababa, found that the spatial growth of the city is mainly characterized by unplanned settlements, which account

for 78% of the total area. Obviously, the majority settlement areas in Addis Ababa are unplanned settlements and are generally characterized by leapfrogging development and often result in fragmentation, which eventually increases the NP. On the other hand, the low PD, SHAPE_MN, ENN_MN, and BC values in low-density mixed residence land use areas compared to medium- and high-density mixed residence areas showed that the UGS in functional land use areas was more consistent than in the medium-density residence and high-density residence areas. This implies that the UGS located in the peri-urban area are relatively less complex than in inner city areas. This finding aligns with the earlier study by You [52], who concluded that the outer districts tend to be less fragmented. However, this is in contrast with [51], who argued that the UGS in outer districts tend to be more fragmented [51].

Similarly, ENN_MN values in areas used for commercial activity, municipal services, and administration functions were high, showing that the UGS in these functional land use areas were isolated and sporadically located. This is due to the fact that highly urbanized areas are more homogenous than others, while low-density areas are heterogeneous [48], as verified by the highest LPI value being shown for commercial areas. In other words, a higher ENN_MN value indicates a higher degree of discretization of UGS patches. Conversely, the degree of discretization of the UGS in the residential area was the lowest, indicating a better connection of individual UGS patches because they were close to each other. This finding is consistent with [48].

This study also identified that the correlation among landscape metrics was strong, while the relationship between landscape metrics and the urban form indices was moderate. For example, SHAPE_MN negatively correlated with BD, while PD positively associated with the BC. Similarly, BD positively associated with PD and NP, indicating that urban form moderately influenced the spatial composition and configuration of the UGS in the study area. Similarly, the relationship between the PARA and the UGS was identified as moderate. The impact of PARA on the UGS composition and configuration was also found to be insignificant, implying that the complexities of building footprints cannot significantly influence UGS patterns, a finding that correlates with earlier studies [46,48]. In contrast to this, Huang and Yang [46] showed that urban form indices such as road networks had a significant impact on the structure of UGS, confirming that urban landscapes dissected by road result in high fragmentation. The variations in the result may be related to the urban form indices selected for the study and the size of the city, i.e., Chinese cities are big, while African cities are relatively small.

While the present study provided some evidence on the composition and configuration of UGS in Addis Ababa, there are some limitations that need to be addressed in future works. First, the satellite imagery used in this study had a 30 m resolution, and with such a medium size resolution all UGS may not be captured, which may have affected the results of the study. Therefore, future studies need to employ very high-resolution (VHR) satellite images to give a better understanding of the UGS spatial pattern characteristics in functional urban land use areas. Second, as the compositions and configurations of UGS markedly vary according to socioeconomic background, future studies should incorporate these variables to give an improved understanding of the UGS characteristics. Third, the building footprints employed for the study did not cover the entire city and might have influenced the correlation analysis between landscape metrics and urban form. Thus, future investigations need to use more updated building footprint data in order to better understand the impacts of urban form on metrics.

## 5. Conclusions

The Ethiopian National Urban Green Infrastructure Standard stipulates that UGS should be provided at a rate of 15 $m^2$/inhabitant in an urban area. Similarly, the new structure plan approved by the Addis Ababa city administration anticipates increasing the per capita UGS from the current 1 $m^2$/inhabitant to 15 $m^2$/inhabitant by the year 2025. Achieving such a high per capita UGS rate in African cities such as Addis Ababa

is extremely challenging, given that the city is rapidly urbanizing and the available UGS are shrinking, as well as the demand for such facilities is outstripping the supply. This is compounded by the current planning practice, which is mainly focused on morphological analysis and relegated UGS planning to a secondary position and exacerbated the problem. Specifically, the current planning approach does not take into account the composition and spatial configuration of UGS in different urban functional land use areas while planning for new developments. Nevertheless, the study has shown that, it is important to understand how the spatial characteristics of UGS affect the social, environmental, and economic benefits, which might be less understood by spatial planners. To this end, this study provides a new insights into how UGS compositions and configurations vary in different functional urban land use areas, as well as the relationships between landscape metrics and urban form, and thus might help to improve the planning practice towards a more sustainable urban environment.

The present study focused on understanding the composition and configuration of UGS in built-up functional land use areas such as residential, commercial, administration, social services, manufacturing and storage, transport, and municipal services areas. The result showed that the high-density mixed residence, medium-density mixed residence, and low-density mixed residence areas contained 16.7%, 8.7%, and 42.6% of the UGS, respectively, and occupied 67.5% of the total UGS in the study area. Meanwhile, manufacturing and storage, transport, social services, administration, municipal functions, and commercial areas accounted for 11.5%, 6.6%, 8.2%, 3.3%, 1.3%, and 1%, respectively, accounting for only 11.5% overall, showing that the largest proportion of UGS were found in residential areas. The study also revealed that 86.2% the UGS were smaller than 3000 $m^2$ and 13.8% were larger than 3000 $m^2$, demonstrating a high level of fragmentation. The results also showed that there was a strong correlation among landscape metrics, while the relationship between urban form and landscape metrics was moderate. For instance, CA exhibited a strong positive correlation with NP (correlation = 0.99 **, $p < 0.001$), while PD was strongly negatively correlated with SHAPE_MN (correlation = $-0.93$ **, $p > 0.001$). Similarly, the nearest neighborhood distance (ENN_MN) negatively correlated with the CA (correlation = $-0.61$). Meanwhile, the relationship between landscape metrics and urban form was found to be fair, demonstrating its moderate impact on the UGS spatial characteristics. The results also revealed that compared to outer sub-cities, inner sub-cities in Addis Ababa had a very low proportion of the UGS, which did not meet either the local or international standards (15 $m^2$ for the local standard and 9 $m^2$ for the WHO standard).

In general, based on a case study of Addis Ababa, Ethiopia, the results showed that the UGS were disproportionately located in residential land use areas, showing that the UGS pattern was influenced by the heterogeneity of urban land use functions. Hence, a set of planning measures need to be implemented to increase the composition and improve the configuration of the UGS. Firstly, urgent urban planning measures should be taken to increase the quantity of UGS in the inner districts. Secondly, future urban planning, design, and management should be guided by an understanding of UGS composition and configuration. Thirdly, urban planning practice must strike a balance between economic development and UGS planning, as the city is currently running out of space. Fourthly, as Addis Ababa is a land-constrained city, the existing spatial planning system needs to shift from the traditional planning approach to a more pragmatic approach and introduce a micro-level UGS planning strategy to transform rapidly deteriorating urban environments, improve livability, and make the city to achieve the Sustainable Development Goals (SDG) set by the UN.

**Author Contributions:** Conceptualization, Methodology, Software, Validation, Formal Analysis, Investigation, Data Curation, Original Draft Preparation, Writing E.M.W.—Supervision, Review and Editing P.V.G. Both authors have read and agreed to the published version of the manuscript.

**Funding:** This research received no external funding.

**Data Availability Statement:** Data will be available up on request. Images employed for the study will be available online for readers.

**Conflicts of Interest:** The authors declare no conflict of interest.

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
