# Peer review of "Urban Green Space Composition and Configuration in Functional Land Use Areas in Addis Ababa, Ethiopia, and Their Relationship with Urban Form"

_land, doi:10.3390/land10010085_

Round 1

Reviewer 1 Report

This an interesting research paper ( and not just a case report, on my opinion) that presents an in-depth study of the spatial composition and configuration of Urban Green Spaces (UGS) in Addis Ababa, Ethiopia, thus contributing to this kind of studies with a case-study in the African context. The research methodology is appropriate and the findings are important. The paper deserves publication since it has a high value both for the analysis undertaken and for the suggestions made concerning possible planning interventions. 

However, I would advice the authors to make a very careful proof reading in English since there are many points in the text that need a rephrasing and re-writing. I have tried to indicate some of these points adding several notes in the text attached. I would also ask the authors to completely re-write their abstract. It is full of numbers ( e.g percentages ) thus failing to give the key messages resulting from the study. I have also picked up some of these key findings in the text that should be on my opinion mentioned in the abstract. 

As a general remark, the article should be simplified as far as the way of presentation is concerned, in order not to be confusing and be more consistent and easily understandable to the readers. 

Reviewer 2 Report

The manuscript “Urban Green Spaces composition and configuration in functional land uses and their relationship with urban form” is a very interesting case study that focuses on the spatial configuration of the green spaces in Addis Ababa city. The topic fits the aims of the journal and is relevant on the discourse of sustainable urban planning. Generally, I think that it is a good paper, the results are interesting but not much clear (see below for further details). The structure of the paper is generally good. Also I am not a native speaker, the paper requires an English revision. Furthermore, there are several typing errors in the text. From the science point, I would not make many changes. I would recommend the publication after MINOR REVISIONS.

Study area:

The authors should better clarify the different used data for urban population.

Fig.1 Inser the total land use for the study area.

Data and methods:

The authors should insert the source and the quality and validity of Functional Urban Land Use (Minimum Mapping Unit and Minimum mapping width).

2.2.3. Urban form metrics

The authors should move the first part of this section on Introduction (from Most recent…. To composition and configuration). Furthermore, for PD index use a unique unit for all part in the text (inhab./km2) and this index is not a ratio!

  1. Results

The authors should insert a summary table (with the Pearson index correlation values) for their statistical analysis. In my opinion, figure 3b is not relevant for the conducted analysis.

Figure 4: Insert the ten sub-cities boundaries and the label name.

Table 4: Replace the numbers with the name of sub-cities.

Table 6: the authors should specify in the text why they have used these area intervals (Cluster analysis? K-means? Other methods?)

Conclusions:

First part just repeats some facts stated already below. The authors should indicate what is new, how they contributed to science and if there can anything else evolve from your findings.

Reviewer 3 Report

The authors did a good job investigating and quantifying the spatial composition and configuration of UGS in Addis Ababa. The manuscript addresses an important topic, especially in the context of urban planning and environmental justice. The value of the article is that there are few similar African case studies in the literature. However, this is also its weakness because it analyses a very special case and contains few general messages to the international audience which was not previously known.

The manuscript has sound methodological approach with various statistical and spatial analysis methods. These are well established and generally used in similar research. I recommend to use Fragscape, which is a plugin to QGIS and can compute effective mesh size, which can add new insights to this analysis.

The article is certainly well written and after minor changes I recommend its acceptance for publication.

Specific comments and suggestions to the authors:

In the abstract, the SHAPE_MN abbreviation is not resolved (unlike the others).

On Figure 1. the smallest auxiliary map is redundant, I recommend to delete it, while the one with the area of Ethiopia can be bigger, and recommend to just show Addis Ababa from the cities (the other cities location is not relevant here).

In subsection 2.2 it would be important to present the latest master plan of the city, which the authors refer in subsection 3.3. This master plan was an important spatial dataset in the analysis.
